# Kidney Proximal Tubule GLUT2—More than Meets the Eye

**DOI:** 10.3390/cells12010094

**Published:** 2022-12-26

**Authors:** Majdoleen Ahmad, Ifat Abramovich, Bella Agranovich, Alina Nemirovski, Eyal Gottlieb, Liad Hinden, Joseph Tam

**Affiliations:** 1Obesity and Metabolism Laboratory, Faculty of Medicine, The Institute for Drug Research, School of Pharmacy, The Hebrew University of Jerusalem, Jerusalem 9112001, Israel; 2Rappaport Faculty of Medicine and Research Institute, Technion, Haifa 3525422, Israel

**Keywords:** KPTCs, GLUT2, mTORC1, AMPK, diabetic kidney disease, SGLT2, CB1R, SREBP1, importin-α1

## Abstract

Tubulopathy plays a central role in the pathophysiology of diabetic kidney disease (DKD). Under diabetic conditions, the kidney proximal tubule cells (KPTCs) are exposed to an extensive amount of nutrients, most notably glucose; these nutrients deteriorate KPTCs function and promote the development and progression of DKD. Recently, the facilitative glucose transporter 2 (GLUT2) in KPTCs has emerged as a central regulator in the pathogenesis of DKD. This has been demonstrated by identifying its specific role in enhancing glucose reabsorption and glucotoxicity, and by deciphering its effect in regulating the expression of the sodium-glucose transporter 2 (SGLT2) in KPTCs. Moreover, reduction/deletion of KPTC-GLUT2 has been recently found to ameliorate DKD, raising the plausible idea of considering it as a therapeutic target against DKD. However, the underlying molecular mechanisms by which GLUT2 exerts its deleterious effects in KPTCs remain vague. Herein, we review the current findings on the proximal tubule GLUT2 biology and function under physiologic conditions, and its involvement in the pathophysiology of DKD. Furthermore, we shed new light on its cellular regulation during diabetic conditions.

## 1. Introduction

Glucose is an important energy source for most human cells [1,2]. It is the ultimate fuel molecule for the brain, and its metabolism is considered the fastest source for ATP production; therefore, blood glucose is kept within a constant and stable range, achieved by a highly dynamic influx and efflux from the cells [3]. The cellular uptake of glucose is mediated by specialized transport proteins (glucose transporters) found on the cellular membrane. These transporters also contribute to glucose cellular metabolism that occurs in the primary metabolic organs, including the intestine, pancreas, liver, and kidneys. The understanding of glucose homeostasis through the different compartments of the body has increased over the years. However, maladaptive glucose metabolism and transport are associated with arguably the most devastating/destructive diseases of our century (such as diabetes and its complications, as well as metabolic syndrome and cancer) [4,5]. In this regard, the knowledge about glucose homeostasis, particularly transport, and the associated molecular pathways under pathological conditions is in its first steps.

To date, fourteen members of the facilitated glucose transporter (GLUT) family have been identified and characterized, as well as another six sodium-glucose cotransporters (SGLTs) [6,7,8]. The expression and distribution of these GLUTs are variable among different cell types, and their functions and substrate affinities are diverse. For comprehensive reviews recapitulating the different GLUTs and their features, see [8,9,10,11,12,13]. Moreover, for emerging information about glucose transport and metabolism in the major metabolic organs in health and disease, the reader is referred to [10,14,15,16,17].

Here, we focus on one of the most vital organs in glucose homeostasis, the kidney [specifically the kidney proximal tubule cells (KPTCs)], and we inclusively summarize the knowledge related to one specific GLUT, GLUT2. Furthermore, we shed new light on its cellular regulation during diabetic conditions. In this way, we aim to provide a clearer picture of the current knowledge, subsequently assisting future research in targeting the missing pieces of our current understanding.

## 2. Kidney Proximal Tubule GLUT2 under Physiological Conditions

Among several functions, the kidneys play a vital role in regulating glucose homeostasis. They are responsible for tightly maintaining constant blood glucose levels (within the normal range). In normal conditions, this function is mediated by the intensive work of ~2 million nephrons, handling about 180 g of glucose daily. One segment, in particular, is of a great importance in the management of glucose homeostasis, namely, the proximal tubules (PTs) composed of KPTCs. These polarized epithelial cells are intended for transport: on the one side, there is a basolateral membrane (BLM) bordering the interstitium and the blood capillaries, whereas on the apical side, there is a brush border membrane (BBM) that increases the cell surface; therefore, improving reabsorption of solutes including glucose [18,19].

The unique KPTCs contribution to glucose homeostasis is mainly related to the processes of glucose reabsorption and its production via gluconeogenesis. The transport apparatus is coordinated functionally and structurally by two steps: First, glucose is transported from the tubular lumen to the intercellular space by an active transport (against the glucose concentration gradient), which is achieved by the SGLT2 in the early proximal tubule (S1/S2 segment) and SGLT1 in the later parts of the proximal tubule (S2/S3 segment). Second, once the glucose that has accumulated inside the cells surpasses the glucose concentration in the interstitial space, an ATP-independent transport (passive) along the concentration gradient is mainly coordinated by the facilitated GLUT2 located on the BLM of KPTCs, and can be further supported by GLUT1 in the distal segment [3,9,10,19,20].

Similar to the “conformational model” suggested for SGLT2 action [21], GLUT2 is a 12-domain transmembrane protein that changes its confirmation upon binding to a glucose molecule. This new shape enables it to release the glucose to the other side [22]. GLUT2 is usually located on the BLM of the epithelial KPTCs; in other circumstances such as hyperglycemia, an increased glomerular filtration rate (GFR) acutely under euglycemia, or increased GFR in chronic kidney disease, it is translocated to the apical BBM (see Section 3).

A growing body of evidence suggests that GLUT2 is a key protein in regulating glucose homeostasis in the proximal cells, both in normal as well as pathological states [10,19,20]. In this section, we review the different roles that have been attributed to GLUT2 in the KPTCs under normoglycemia.

### 2.1. Role of KPTC-GLUT2 in Glucose Reabsorption

Various studies reported a difference in the reabsorption capacity of the different segments along the PT, indicating that the bulk of glucose (>90%) is reabsorbed almost entirely in the early segments (S1, S2), and only the remaining minority (~10%) is rescued by the later PT (S3). Interestingly, this feature is highly correlated with the unique expression and distribution of different glucose transporters along the PT (Figure 1). Whereas SGLT2 and GLUT2 are predominantly expressed in S1 and S2, SGLT1 and GLUT1 are mainly abundant in the S3 segment [20,23]. Interestingly though, the spatial transcriptomics reveals a greater abundance of GLUT1 also in the medullary part of the kidney (Figure 1). Therefore, this indicates that the absorption capacity results from the synchronized work between active and passive glucose transporters (SGLT2 and GLUT2, respectively) with optimized kinetic features. Indeed, the low affinity (high Km) and high capacity features of GLUT2, besides its ultimate expression in the early segments of the tubular system, are essential to match the proper functionality of the low affinity and high capacity SGLT2 transporter [24,25].

The unique kinetic features of GLUT2, among other GLUTs, emphasize GLUT2’s ultimate role in the transport process. Glucose that enters the KPTCs is accumulated until its concentration reaches around 15–20 mmol/L (GLUT2 *Km* to glucose), consequently permitting its diffusion by GLUT2 from the intercellular space to the interstitial space [18,19]. In the interstitium, glucose is either returned to the circulation via peritubular capillaries or used for anaerobic glycolysis (for energy/ATP production) in distal parts of the nephron. Hence, GLUT2 in the KPTCs greatly contributes to fine-tuning a precise glucose equilibrium in the bloodstream.

Since the main mission is to recover maximum glucose and prevent the loss of this valuable energy source, most of the reabsorption process is coordinated by insulin independent transporters (GLUT2, SGLT2) [3]. Confirmation of the significance of GLUT2 in glucose reabsorption was elucidated in murine and human studies showing that the functional loss of GLUT2, either by a genetic knockout of the *SLC2A2* gene in vivo or mutations in humans [Fanconi- Bickel Syndrome (FBS)], led to glucosuria [20,26,27] (see Section 3). These results indicate that GLUT2 malfunction in the KPTCs not only affects the diffusion of glucose to the interstitium—it also plays a primary role in regulating the reabsorption on the luminal side, although the mechanism underlying its regulation has not yet been clarified.

This observation was further strengthened by utilizing micro positron emission tomography (micro-PET) studies. Using this non-invasive imaging technology, the Wright group injected 2-FDG (a GLUT2 tracer) and Me4-FDG (a SGLT substrate with a low affinity for GLUT2) into wild type (WT) and GLUT2^−/−^ mice, followed by measuring the clearance of the different PET tracers. In their work, injecting Me4-FDG showed no measurable excretion in the urine of the WT mice; on the other hand, it appeared in the urine of the GLUT2^−/−^ animals [28]. Their results impose several options: first, GLUT2 directly participates in the transport of glucose from the apical side (it is expressed on the apical membrane). Second, GLUT2 indirectly takes part in glucose transport by regulating the SGLT2 expression; therefore, this explains how nullification of GLUT2 affects Me4-FDG reabsorption. Finally, deletion of GLUT2 may affect SGLT2 expression and/or function indirectly, probably via regulating the Na^+^ gradient or through other pathways. The former and latter possibilities have not been reported, whereas GLUT2 regulation of SGLT2 was suggested recently (discussed in Section 3). It is clear though, from the above-mentioned studies, that GLUT2 in the KPTCs is necessary for renal glucose reabsorption.

### 2.2. KPTCs-GLUT2’s Role in Gluconeogenesis

Outside the exclusive role of glucose reabsorption, KPTCs contribute to glucose homeostasis by their unique ability to produce glucose via gluconeogenesis (GNG); ~20–50% of whole-body endogenous glucose production [29]. Specific cells in the kidney (mainly in the S1 and S2 segments) are specialized in the endogenous production of glucose, whereas cells in the S3 segment were found to utilize glucose via glycolysis [14].

The special metabolic pathway of GNG produces glucose from non-carbohydrate substrates such as lactate, glutamine, and glycerol. In the KPTCs, the preferable source for GNG is the amino acid glutamine [30], whereas lactate is the most predominant substrate for both hepatic and renal GNG [3,19]. It is important to note that there are differences in the substrates used in the different segments of the proximal tubule. Concurrently, GLUT2 (and SGLT2) activity within the highly active segments (S1 and S2) (Figure 1) affects the availability and concentration of the GNG substrates. For instance, glutamine is the preferred source in the S1, compared to lactate in the S2 and S3 segments. This is also related to the O_2_ gradient across the kidney (more GNG in the oxygenized cortex versus more anaerobic glycolysis in the hypoxic medulla). Other factors include a rich mitochondrial content in the early segments compared to the late PT.

Following GNG, the glucose formed is released into the interstitium by GLUT2. Glucose produced in the gluconeogenic process mainly contributes to the systemic glucose levels and to a lesser extent to the more distal parts of the nephron for energy production by glycolysis [25]. Interestingly, despite an extensive glucose flow through the PT, the KPTCs themselves do not prefer to use glucose for energy generation [10], but mainly metabolize fatty acids, glutamate, lactate, and ketone bodies as energy sources [31,32]; therefore, one should discriminate between their metabolic needs and their functional role in regulating glucose homeostasis. Moreover, glucose transport in the KPTCs is thought to work together with GNG to prevent an overload of glucose (glucotoxicity). Still, it is unclear whether activating the transport system downregulates GNG or vice versa, due to studies showing both cases. More research is needed to address this question and further assess the role of GLUT2 in mediating this synchronization under normal conditions [10].

Generally, the KPTCs greatly contribute to acid-base homeostasis by participating in the reabsorption of various electrolytes. In fact, “de novo” bicarbonate generation from glutamine metabolism links glucose transport and acid-base state regulation. GNG, which occurs in these cells, also plays an important role in this regulation, specifically by promoting the production of bicarbonate as a by-product [10]. In the fasting state/starvation (post-absorptive phase), KPTCs account for 40–50% of total glucose production in the body, in comparison with their contribution in the postprandial phase (~10%). Additionally, during acid overload (metabolic acidosis), up to 20% of the total glucose production is provided by the kidney. The latter takes place primarily in the S1 and S2 segments of the PT, where GLUT2 expression is abundant. Interestingly, a general GLUT2 loss in humans is characterized by renal tubular acidosis, indicating a link between GLUT2 and maintaining a healthy acid-base balance. However, more research is needed to validate and dissect this link [13]. To conclude, it appears that coordination exists between glucose transport, GNG, and catabolism (utilization). New studies should focus on understanding the link between all these processes.

## 3. Kidney Proximal Tubule GLUT2 under Pathological Conditions

An increased value for the contribution of GLUT2 is added in disease states. All the above-mentioned functions and processes can be altered under pathological conditions, thus affecting and altering the expression and function of GLUT2.

### 3.1. Genetic Mutations in the SLC2A2 Gene (GLUT2)

In humans, inactivating mutations in the GLUT2 gene (*SLC2A2*) cause FBS. FBS is a fatal rare autosomal recessive disorder with a complex phenotype, mainly due to GLUT2 expression in several metabolic organs, including the intestine, pancreas, liver, and KPTCs. Major renal characteristics include tubulopathy, characterized by glycosuria, phosphaturia, aminoaciduria, proteinuria, and hyperuricemia [13,26,33,34,35]. Other more severe symptoms include growing problems during infancy, rickets, in addition to hepatomegaly and enlargement of the kidney due to glycogen buildup [10]. The glycogen accumulation could be the reason behind the renal tubulopathy observed in these patients. In addition, GLUT2 knockout mice exhibit decreased expression of the Na^+^-phosphate type 2 transporter (NPT2C), suggesting that GLUT2 expression regulates the expression of other transporters, leading to the observed phosphaturia [9]. Moreover, glucosuria, among the remaining symptoms, could result from abnormal glucose homeostasis caused by defects in liver glucose handling, insulin/hormonal imbalance, and impaired glucose management in each of the associated metabolic organs.

On the other hand, recent studies showed that local inhibition, downregulation, or deletion of GLUT2 specifically in the kidney or in the KPTCs protects the kidney from the deleterious effects of diabetes and hyperglycemia [36,37,38]. These results raise various questions: Are there other GLUTs working under pathological conditions? Is there one or more alternative compensatory mechanisms for KPTCs-GLUT2 loss under pathological conditions? Alternatively, do the FBS renal phenotypes result from the systemic loss of GLUT2, or directly result from GLUT2 absence in the KPTCs? The aforementioned genetic conditions, including FBS, further imply a regulatory role for GLUT2 in glucose homeostasis in the body, in general, and in the KPTCs, in particular.

### 3.2. KPTC-GLUT2 in Diabetic Kidney Disease

Owing to the major role played by the KPTCs in regulating glucose homeostasis, and the fact that glucose entry into these cells is insulin independent, makes them more susceptible to altered metabolism under hyperglycemic conditions. Diabetic kidney disease (DKD) is a central complication of diabetes. In diabetes, high blood glucose levels and a wide range of systemic alterations affect the organs’ ability to uptake and utilize glucose. Furthermore, pathologic processes such as inflammation and fibrosis are emerging. Hyperglycemia leads to the development of tubulopathy, among other structural and functional changes in the renal tissue. On the other hand, maladaptive glucose regulation in the PTs has immense effects on systemic glucose balance and metabolism; therefore, it is associated with deteriorating nephropathy. It is not surprising then, that DKD raises the risk for other cardiovascular diseases, affecting between 30% and 50% of diabetic patients (T1D and T2D, respectively), among other complications. Despite increasing research aiming to delineate the DKD cellular mechanisms, more efforts are required to obtain sufficient knowledge and targets for therapeutic intervention.

In DKD, glucose transport, metabolism, production, and usage are altered [39]. Moreover, there is inconsistent data in each of the abovementioned fields, consequently confirming our shortage of appropriate solutions. In general, the contribution of GLUT2 to the pathogenesis of the kidney is poorly understood. Yet, following the evaluation and clinical usage of the promising SGLT2 inhibitors, more interest has been invested in studying the role and function of KPTC-GLUT2 in DKD pathophysiology. This interest has raised some exciting and important insights into the field, which will be further discussed in the next subsections.

#### 3.2.1. Glucose Transport/Reabsorption under Hyperglycemia

As discussed earlier, the KPTCs sustain a finite glucose balance by reabsorbing almost all filtered glucose. Under hyperglycemia, the glucose concentration exceeds the capacity of the transporters (T_max_ of glucose transporters), resulting in glucosuria [3]. The glucose filtration load, which is determined by two main factors, GFR and blood glucose levels, is magnified in diabetic patients [GFR is increased in the early phase of diabetes [9,10]]. Consequently, the KPTC reabsorption rate and capacity are enlarged and can reach 23.3 mmol/min (compared to 19.5 mmol/min in healthy individuals) [10,19]. This increase is strongly linked to changes in the transport KPTCs system, including alterations in the expression level, activity, and localization of the glucose transporters.

Alterations in the expression of GLUT2 have been reported in different organs during diabetes, and are further implied in the kidney. It is known that glucose transport in the KPTCs is increased in diabetes; however, reports on the expression levels of the glucose transporters are more variable. In the KPTCs, many studies have shown increased protein and mRNA expression of GLUT2 (Table 1 summarizes the expression levels of GLUT2 in the KPTCs under hyperglycemia). The regulation of GLUT2 expression is thought to be mediated by hepatocyte nuclear factor (HNF)-1α, (HNF)-4α, (HNF)-3β [40,41], or sterol regulatory element-binding protein 1 (SREBP1) [37] in the KPTCs.

#### 3.2.2. KPTC-GLUT2 Translocation under Hyperglycemia

Following the discovery of the apical GLUT2 recruitment by SGLT1 in the intestine [54,55,56], GLUT2 translocation to the apical/luminal BBM side was also reported [49]. It was further validated by other models [50,57] as well as by us [36]. In fact, Marks et al., found that in diabetic rats (streptozotocin (STZ)-induced T1D) facilitative glucose transport is increased at the BBM by 67.5%, mainly due to GLUT2 translocation [49].

According to the proposed mechanism in the intestine, glucose absorption via SGLT1 triggers membrane depolarization and Ca^2+^ influx, resulting in cytoskeletal rearrangement, apical localization, and activation of protein kinase C βII (PKC βII), which in turn, mediates GLUT2 apical translocation [54,55,56]. We and others have shown that the apical trans-localization of KPTCs GLUT2 during diabetes is associated with PKC βI activation [36,50]. In our study, we found that this effect is highly regulated by the cannabinoid-1 receptor (CB_1_R) [36,37]. In fact, we have recently demonstrated that genetic deletion or pharmacological blockade of KPTCs CB_1_R ameliorates diabetes-induced renal dysfunction, inflammation, and fibrosis, which goes hand in hand with GLUT2 downregulation [36,37]. Similar to the recruitment of GLUT2 by SGLT1 in the intestine, is it possible that SGLT2 in the PTs recruits GLUT2 under hyperglycemic conditions? Although it has not reported yet, a regulatory effect in the opposite direction has been recently suggested [37,38,58]. We and others showed a correlation between GLUT2 levels and SGLT2 expression in the KPTCs regardless of the glycemic condition [37,38]. Furthermore, Umino and colleagues suggested that activation of the GLUT2/importin-α1/HNF-1α pathway by basolateral high glucose levels enhances SGLT2 expression in KPTCs, and that GLUT2 silencing blocks this increased expression [58]. Deciphering the link between these two transporters under hyperglycemia has a magnificent impact, especially for pharmaceutical development. In other words, SGLT2 inhibitors might induce decreased glucose absorption, partially due to reduced GLUT2 insertion into the apical BBM.

There are several possibilities for GLUT2 translocation from the BLM to the BBM of the KPTCs: First, translocation supports active transport by SGLT2 that cannot withstand the overload of glucose (a compensatory-deteriorative effect) [49,57]. Second, GLUT2 is recruited to the apical side to act as a glucose sensor (as suggested in other organs). In this regard, the role of KPTCs-GLUT2 in mediating intercellular signaling has been under-researched. Third, owing to increased glucose accumulation in the cells, GLUT2 is recruited to the apical membrane to work as a facilitated transporter; it leaks glucose back to the lumen in order to decrease glucotoxicity. Recently, it was hypothesized that if this is the case, leaking glucose may accelerate apical recycling of glucose and therefore enhance sodium reabsorption through SGLTs [10]. Such a suggestion might have both advantages and disadvantages, since it might aid the machinery of an active transport (by SGLT2), which is working intensively under hyperglycemic conditions, or enhance the Na^+^ supply for the SGLT2 transporters to increase their activity; hence, it leads to worsening the hyperactivity of these transporters. Finally, the rise in blood glucose levels may hamper the tubule-to-blood glucose gradient, essential for glucose flux through BLM GLUT2 [43]. Therefore, GLUT2 recruitment to the BBM may assist in glucose transfer into the KPTCs to maintain a “new homeostatic” tubular glucose gradient.

#### 3.2.3. Glucose Formation (Gluconeogenesis) under Hyperglycemia

Generally, GNG is initiated for glucose formation under situations of low glucose levels (e.g., fasting and starvation). Despite this logic, in diabetes, GNG in the kidney and liver is increased (both in T1D and T2D). Total glucose formation may be amplified by 300% of the normal range in T2D patients. Although inconsistent with the liver, GNG in the KPTCs is stimulated after glucose ingestion (Postprandial phase) by almost 2-fold [39].

Studies in humans as well as in diabetic animal models demonstrated an over-release of glucose into the bloodstream upon fasting [59,60,61], probably because of two contributors: absorption and production. Until now, most attention has been devoted to inhibiting glucose absorption mainly by targeting SGLT2; therefore, investigating the inhibition of other transporters such as GLUT2 is valuable. Furthermore, it is reasonable to evaluate inhibiting GNG/endogenous glucose production in diabetes as another therapeutic approach. Supporting studies for both rationales have been reported [37,38,62].

The maladaptive gluconeogenic activity in the KPTCs ameliorates the elevated blood glucose levels, not only in the postabsorptive phase (fasting glucose) but also in postprandial glucose, perhaps further deteriorating hyperglycemia in diabetic patients. The unpredicted increase in renal GNG upon glucose ingestion is suggested to enhance the refilling of glycogen stores in the liver [29,63]. Additionally, elevated GNG in the diabetic kidney is proposed to take part in several metabolic changes, such as fueling the distal regions of the nephron [10]. In particular, hyperfiltration and enhanced active transport in diabetes raise the energetic needs of the nephron. Accordingly, glucose supplementation to other distal parts of the nephron improves the kidney’s management of energy consumption/utilization. However, greater glucose reabsorption in parallel with increased glucose formation worsens hyperglycemia [10].

The reasons behind the increased GNG in hyperglycemia include insulin regulation, the availability/abundance of GNG precursors, and increased levels of free fatty acids (FFAs). Following our discussion on GLUT2’s role in GNG under normal conditions (see Section 2.2), and since GLUT2 is the main transporter accountable for glucose release upon GNG, what is the link between GLUT2 and GNG under hyperglycemia? How is their activity synchronized, and adapted under pathological conditions? Certainly, the role of renal GLUT2 in regulating GNG or vice versa requires further research.

#### 3.2.4. Glucose Metabolism via Other Pathways under Hyperglycemia

Other metabolic pathways involving glucose include glycolysis and glycogen polymerization. As previously mentioned, the high energetic demands of the KPTCs are mostly achieved by catabolizing FFAs rather than through glycolysis, because they provide a higher ATP yield. A developed transport system works to maintain the homeostasis of different solutes by ATP-dependent transport; these cells are occupied with high concentrations of mitochondria, besides supplying sufficient O_2_, all together, they support an aerobic ATP production over the anaerobic glycolysis. Glycolysis, on the other hand, is the primary process for ATP production in more distal parts of the nephron (including the S3 segment of KPTCs). Kidney glycolysis is altered in diabetes [31,32,33,34], yet no one has reported a link between renal GLUT2 activity and glycolysis under DKD. Meanwhile, genetic inhibition of hepatic GLUT2 has been shown to alter glycolytic and lipogenic gene expression [64,65], further suggesting that the metabolic shift occurring in diabetic conditions may be very well governed, at least in part, via GLUT2.

In normal conditions, a few glycogen stores are accumulated in the kidney; previous studies indicated that the glucose produced or reabsorbed in the KPTCs is consequently released by the basolateral GLUT2, and that it is usually kept as free glucose instead of glycogen polymers. Accordingly, glucose is either returned to circulation or used for anaerobic glycolysis in the hypoxic distal parts of the nephron. However, under diabetic conditions, an excess of glycogen deposition in tubular epithelial cells is observed [66]. This characteristic is usually explained by the overall increased glucose uptake, still the GLUT2 relationship to glycogen polymerization has not been evaluated under diabetic conditions. This is especially important because GLUT2 plays an alternative role in glucose uptake regulation under diabetic conditions. Similarly, the effect of GLUT2 knockout on kidney glycogen polymerization in normal and diabetic conditions needs to be examined.

#### 3.2.5. KPTC-GLUT2’s Role in Regulating Pathological Pathways under Hyperglycemia

To examine the specific role of GLUT2 in KPTCs in the development of DKD, we have recently developed and characterized a specific KPTC-GLUT2-KO mouse strain by using a Cre*^SGLT2^*-Lox system under the Akita diabetic background (for detailed information, see [37]). Interestingly, the reduced GLUT2 expression in KPTCs was sufficient to protect diabetic mice from developing DKD by specifically reducing glucose reabsorption and increasing glucosuria [37]. These null mice were completely protected from the diabetes-induced renal dysfunction, inflammation, and fibrosis, reiterating the importance of KPTCs GLUT2 in DKD pathogenesis.

A recent study by de Souza Cordeiro and colleagues [38] confirmed these results. By utilizing the tamoxifen-inducible Cre*^ERT2^*-Lox system to generate a whole-kidney GLUT2 knockout mouse strain, the authors demonstrated that exposing these mice and their WT controls to STZ or a high-fat diet (mimicking T1D or T2D, respectively) induces glycosuria, enhances urinary volume, and improves systemic glucose intolerance [38]. Taken together, these findings further imply renal GLUT2 as an important regulator of kidney glucose homeostasis, DKD pathogenesis, and whole-body energy metabolism.

#### 3.2.6. KPTC-GLUT2’s Role in Glucose Sensing under Hyperglycemia

The regulatory role of GLUT2 in modulating glucose homeostasis and glucose-sensitive gene expression has been widely demonstrated in pancreatic beta cells, liver, and in the central nervous system (reviewed in [9]). Guillemain et al., were the first to show that hepatocyte GLUT2 may transduce a downstream cellular signal from the plasma membrane to the nucleus via its large intracytoplasmic loop [67]. This glucose sensing, tightly coupled to a sustained glucose metabolism, was suggested to be mediated via tethered karyopherins/importins, which transmit a signal to the nucleus to regulate glucose-sensitive gene expression [68]. In fact, importin-α was shown to accumulate in the cytoplasm of glucose-stimulated hepatoma and pancreatic cell lines [69]. Moreover, importin-α, strongly expressed in the proximal and distal tubules, was shown to be increased in diabetic rat kidneys [70], further suggesting that GLUT2 may also sense and signal to the nucleus in KPTCs.

Overall, the aforementioned research outlines the significance and necessity of understanding glucose regulation/management, and GLUT2-associated pathways in the KPTCs within health and disease.

## 4. Reinvestigating GLUT2-Associated Pathways in Akita Diabetic Mice

To shed more light on the role of KPTC-GLUT2 in diabetic conditions, we utilized metabolomics profiling analysis as an approach to decipher the cellular pathways and chemical fingerprints that are associated with GLUT2 activity. We compared the metabolomic profiles of kidney lysates from Akita-KPTC^GLUT2+/+^ mice vs. Akita-KPTC^GLUT2−/−^ animals (see [37]).

### 4.1. Kidney Metabolomic Analysis of Diabetic Akita-KPTC^GLUT2−/−^ Mice

Of the 284 metabolites that were identified in the samples, only 11 were significantly changed (*p* ≤ 0.05) (Figure 2A,B). The abundance of 10 metabolites was increased; four of the most altered metabolites are 5-aminoimidazole-4-carboxamide ribonucleotide (AICAR), dihydroxyacetone phosphate (DHAP), riboflavin, and allantoin, whereas cytidine monophosphate (CMP) was the only metabolite to exhibit a decreased abundance (Figure 2A,B; Appendix A).

### 4.2. AICAR Activates AMPK and Restores Fat Oxidation in KPTC-GLUT2 Null Akita Diabetic Mice

AICAR, an intermediate metabolite in the purine de novo synthesis pathway [71], is a known activator of the AMP-dependent protein kinase (AMPK) [72], which is a key energetic sensor that restores energy homeostasis by reducing energy consumption and increasing energy production (ATP synthesis) [32]. As mentioned earlier, during diabetic conditions, a metabolic shift occurs in KPTCs, changing their energy utilization from fatty acid oxidation toward glycolysis [32]. This transition is manifested by the increased expression of glycolytic enzymes [73,74,75], lipid accumulation [73], which together, might result in tubular atrophy and accelerated interstitial fibrosis [76]. Under these conditions, AMPK phosphorylation is known to be downregulated, consequently reducing fat oxidation and promoting lipogenesis [77,78,79,80,81,82]. Interestingly, AICAR upregulation in our diabetic KPTC-GLUT2 null mice was accompanied by enhanced AMPK activation (measured by phosphorylated AMPK; pAMPK; Figure 3A–C). Examination of the β-oxidation rate-limiting enzyme, carnitine palmitoyltransferase I (CPT1), revealed upregulated gene and protein expression levels (Figure 3D–F). In addition, assessment of key fatty acid β-oxidation enzymes revealed an increased transcription of *Acadm*, *Hadh*, and *Acaa2*, besides a slight but not significant upregulation of *Echs1* (Figure 3G–J). All together, these findings indicate an improved mitochondrial function and restored fatty acid β-oxidation as the main energy source of KPTCs in the Akita-KPTC^GLUT2−/−^ mice.

In the diabetic kidney, elevated levels of the hypoxia-inducible factor (HIF)1α is known to suppress fatty acid β-oxidation [83,84,85]. Interestingly, reduced expression of this key transcription factor was observed in the Akita-KPTC^GLUT2−/−^ mice (Figure 3K), further strengthening the previous observations. Moreover, Akita-KPTC^GLUT2−/−^ mice exhibited significantly reduced levels of CMP (Figure 2A,B and Figure 3L), which mainly originates from RNA degradation [86,87], as well as a robust increase in riboflavin (Figure 2A,B and Figure 3M), a precursor of the two main coenzymes, flavin mononucleotide (FMN) and flavin adenine dinucleotide (FAD). These coenzymes are involved in energy metabolism (reviewed in [88]), and work as cofactors for flavoproteins that play vital roles in the mitochondrial electron transport chain, β-oxidation, as well as mitochondrial function. This may further imply a recovery of the metabolic programing and a return to normal protein translation in the kidneys of diabetic mice lacking GLUT2 in KPTCs.

### 4.3. Reduced Endocannabinoid ‘Tone’ in KPTC-GLUT2 Null Akita Diabetic Mice

AMPK is known to be negatively regulated by KPTC-CB_1_R [81], the furthermost prominent receptor of the endocannabinoid system (ECS) in the diabetic kidney. In fact, ECS activity has been proven to promote pathogenesis and fibrosis in the kidney during hyperglycemia. Alternatively, its suppression/inhibition was suggested as a therapeutic approach for chronic kidney disease by our group [81]. We have shown that specific CB_1_R blockade activates AMPK, resulting in reduced KPTC lipotoxicity and obesity-induced renal dysfunction, inflammation, and fibrosis [81]. Interestingly, the renal levels of the main endocannabinoids known to activate CB_1_R, anandamide (AEA), and 2-arachydonoylglycerol (2-AG) were found to be downregulated in the kidneys of Akita-KPTC^GLUT2−/−^ mice (Figure 4A,E). These findings are most likely mediated via a significant decrease in AEA synthesizing enzyme, *N*-acyl phosphatidylethanolamine phospholipase D (*Napepld*) (Figure 4B) but not its degrading enzyme, fatty acid amide hydrolase (*Faah*), whose expression (Figure 4C) or activity (Figure 4D) was not changed. Moreover, the decreased levels of 2-AG are most likely attributed to the significant increase in 2-AG degrading enzyme, monoacylglycerol lipase (*Mgll*) (Figure 4F) but not its synthesizing enzymes, diacylglycerol lipase (*Dagla/b*), whose expression levels were not affected (Figure 4G,H).

In the light of these results, we assessed the CB_1_R expression levels in our model, and surprisingly, no changes in its gene and protein expression levels in the kidney of the null mice were found (Figure 4I–K). These results support either a regulatory feedback loop between AMPK and the endocannabinoid/CB_1_R system or a link between GLUT2 and the synthesizing or metabolizing enzymes responsible for the 2-AG and AEA levels in KPTCs. Anyway, AMPK activation, along with endocannabinoid ‘tone’ reduction in the Akita-KPTC^GLUT2−/−^ mice, could explain the improved renal alterations (including reduced fibrosis and kidney injury), which were observed in these mice and were reported earlier [37]. Of note, the renal gene expression levels of CB_2_R were upregulated in Akita-KPTC^GLUT2−/−^ mice (Figure 4L), findings that may further explain the reduction in kidney inflammation in these mice [37].

Recently, we have reported that 2-AG production is upregulated under hyperglycemic conditions in KPTCs, consequently enhancing CB_1_R cellular signaling, which further upregulates GLUT2 expression and translocation and increases glucotoxicity [37]. GLUT2 nullification in the diabetic mice resulted in SGLT2 downregulation, decreased glucose reabsorption, and reduced glucotoxicity. However, the correlations between GLUT2 expression and 2-AG production would need to be further evaluated in future studies.

### 4.4. DHAP Upregulation and mTORC1 Activation in KPTC-GLUT2 Null Akita Diabetic Mice

Following the metabolic shift documented in Akita-KPTC^GLUT2−/−^ mice, and in alignment with the elevated fatty acid β-oxidation, we were interested in examining any changes in glucose catabolism/anabolism caused by AMPK activation. Although our metabolomic analysis did not reveal any significant changes in metabolites associated with these pathways (see the Appendix A), there was a significant upregulation of DHAP (Figure 2A,B and Figure 5A). DHAP is known to activate the mammalian target of rapamycin complex 1 (mTORC1) [89,90], a central hub integrating signals from growth factors, nutrients, and energy to promote cell growth, proliferation, and energy storage [32]. Interestingly, DHAP upregulation in Akita-KPTC^GLUT2−/−^ mice was accompanied by enhanced mTORC1 activation, manifested by the enhanced ribosomal S6 phosphorylation (pS6) downstream of mTORC1 (Figure 5B,C). These findings are very surprising, especially in light of the fact that the AMPK and mTORC1 signaling pathways negatively regulate one another [91,92], and that the renal endocannabinoid/CB_1_R system is a positive regulator of mTORC1 [37]. In addition, mTORC1 is known to positively regulate SREBP1 [37,93,94,95], which controls the transcription of glycolysis and lipogenesis-related genes [96,97,98,99,100]. SREBP1 is also a prominent transcription factor of GLUT2, both in hepatocytes [101] and KPTCs [37] under hyperglycemic conditions. Therefore, the over-activation of mTORC1 in the diabetic GLUT2 KO mice reported herein may imply a compensatory effect driven by the lack of GLUT2 and the inability to transcribe it. Thus, continuous research is required to address these unpredicted results.

Another interesting metabolite that was significantly increased in the kidney of Akita-KPTC^GLUT2−/−^ mice is allantoin (Figure 2A,B and Figure 5D), a non-enzymatic oxidation product of uric acid, which is a biomarker for oxidative stress [102] that acts as a reactive oxygen species scavenger [103]. It is found in urine and blood samples of diabetic animals [104,105,106] as well as in polycystic kidney disease patients [107]. Imbalance redox and mitochondrial dysfunction are known features of DKD [108]. Oxidative stress normally suppresses mTORC1 activity [109], and vice versa; mTORC1 is considered an important suppressor of mitochondrial oxidative stress [110,111]. However, overactivation of mTORC1 increases oxidative stress in DKD [112], and as in our case, mTORC1 overactivation could be the reason behind increased allantoin levels found in Akita-KPTC^GLUT2−/−^ mice. However, the latter assumption needs to be further examined. Overall, along with the beneficial effects of restoring renal metabolic homeostasis via the activation of AMPK lies the potential harmful effect of mTORC1 overactivation, which may chronically impair kidney function.

## 5. Targeting GLUT2 in the Kidney—Rationale and Limitations

Understanding the changes and adaptations of kidney cells under pathological conditions is based on identifying new targets, and developing new therapeutic solutions, and/or intervention approaches for treatment, and hopefully the prevention of kidney diseases.

Currently, beyond the great interest in researching SGLT2 inhibitors, kidney glucose transport and metabolism are poorly understood under pathological conditions. In addition, as previously reviewed, there are different opposing results. This is primarily because the current system models are limited and the kidney cells’ environment is hard to mimic/transcribe. Nevertheless, new methodologies and technological advances, such as the evolving omics studies, should help in raising novel discoveries.

### 5.1. The GLUT2-SGLT2 Link

In accordance with the abovementioned suggested role of GLUT2, it has increasingly been more suspected to participate in cellular regulatory cascades in the diabetic kidney, beyond just being a glucose transporter in the reabsorption process. We and others showed a correlation between GLUT2 levels and SGLT2 expression in the KPTCs [37,38,58]. High glucose levels on the basolateral side of KPTCs was recently suggested as an activator of the GLUT2/importin-α1/HNF-1α pathway, leading to increased SGLT2 expression on the apical side of the KPTCs. In this study, GLUT2 inhibition blocks this increased expression [58].

These documented observations have been recently supported by findings from our group and from de Souza Cordeiro et al., demonstrating that KPTC-GLUT2 nullification reduces SGLT2 expression levels, regardless of the glycemic condition [37,38], possibly via the transcription factor HNF-1α [38]. Nevertheless, reduced KPTC-SGLT2 levels have been observed in diabetic conditions in vivo [36,38], in contrast to Umino et al., who showed the hyperglycemic-dependent upregulation of SGLT2 in vitro [58]. These conflicting results may be explained by the different models used and/or the transition from an acute to a chronic phase. Furthermore, in normoglycemic conditions, inhibition of SGLT2 allows GLUT2 to transport glucose from the interstitial space to the intracellular space to equilibrate intra- and extra-cellular glucose levels [113].

In addition, considering the potential regulatory role of SGLT2 on GLUT2 expression under hyperglycemic conditions, SGLT2 inhibitors might induce decreased glucose absorption, partially due to reduced GLUT2 insertion into the apical BBM. It could also postulate that GLUT2 nullification may cause glucose retention in the KPTCs, which in turn, results in reduced SGLT2 activity. Investigating the link between these two transporters under hyperglycemia is especially important for pharmaceutical advancement.

### 5.2. KPTC-GLUT2: Is It Targetable?

The development of drugs modulating GLUT2 to treat diabetes has not been practical, owing to several reasons: First, because of the close structural and functional similarity between the GLUT family members (specifically, the class I GLUT transporters, including GLUT1, GLUT3, and GLUT4) [114,115]. Interestingly a recent study has been used in silico ligand screening to identify a novel selective GLUT2 inhibitor [116]. Using eukaryotic systems, Schmidl et al., validated the GLUT2 specific inhibitors and suggested them for future research and crystallization of GLUT2, as well as other unknown GLUT structures. Such discoveries have the potential to enhance our understanding of GLUT2 function and biology, mainly via providing better tools for new research. However, it is worth considering that investigating such molecules within in vitro models of DKD is limited. For example, immortalized cell lines with tumorigenic features are among the most convenient in vitro models; however, GLUT inhibitors have already shown an ameliorative effect on cancer cells’ phenotype [117]. In other words, it will be hard to dissociate the beneficial effect of GLUT2 inhibitors in DKD models from their positive effect on carcinogenesis.

Second, GLUT2 plays a vital physiological function throughout the body. Thus, it would be challenging to specifically target GLUT2 in the kidney without affecting other organs, such as the intestine, pancreas, brain, and liver, whose blocking may render systemic side effects. In this regard, a recent study has examined the cognitive effects of a general GLUT2 knockout, and found no difference in neither the neurological functions, nor the cognitive abilities of the GLUT2-KO mice, compared to the control [118]. Nonetheless, the novel idea of targeting KPTCs-GLUT2 as a potential therapy for DKD may be very tempting, due to the recent results demonstrating that its nullification in KPTCs/whole-kidney has the potential to protect mice from DKD [37,38]. For that reason, more research that examines the effectiveness of KPTCs-GLUT2 as a pharmacological target is needed.

Recent advances in specific drug delivery systems to the kidney (reviewed in [119]), utilizing nanoparticle-platforms as a tool to deliver therapeutics directly to the damaged kidney and minimizing their systemic effect, may be an appealing strategy to use GLUT2 inhibitors as therapeutics against DKD. Drug delivery systems aiming for the PTs have to overcome the glomerular filtration barrier size and charge [120] and avoid serum protein adsorption [121]. Moreover, the drug delivery systems aim to target KPTCs-GLUT2 should be suitable for targeting the membrane proteins, and not be endocytosed into the tubular cells through the megalin/cubilin-mediated endocytosis, unless the cytoplasmatic machinery responsible for the apical translocation of GLUT2 is targeted.

In parallel, it would be optional to target upstream regulators of GLUT2, such as CB_1_R or downstream effectors, such as importin-α1 or HNF-1α for therapeutic benefits. Providing additional novel information regarding the exact mechanisms by which KPTCs-GLUT2 functions under diabetic conditions has the potential to be translated into relevant therapeutics for treating DKD.

## 6. Gaps and Conclusions

This review raises numerus unresolved questions as well as theoretical and experimental gaps: KPTCs-GLUT2 has an obligatory role in normal renal glucose reabsorption. However, does GLUT2 have a direct effect (by regulating mRNA or protein levels) or indirect effect on SGLT2 expression? Does it play a significant role in the balance between GNG, glycolysis, and glucose absorption? Or does it contribute to maintaining healthy acid-base balance?

KPTCs-GLUT2 nullification was found beneficial in ameliorating DKD. However, are there any compensatory effects or mechanisms for the absence of GLUT2? Does GLUT2 play a role in GNG under hyperglycemic conditions? Does it affect SGLT2 expression/activity? How does GLUT2 nullification affect mTORC1 signaling? Does its nullification induce long-term negative consequences? How does GLUT2 nullification affect the ECS and kidney metabolome? Finally, is it feasible to target KPTCs-GLUT2 for therapeutic purposes? These open questions raise the gaps in knowledge from one hand, but also highlight the great importance and relevance of this research field on the other hand.

Nevertheless, it seems that GLUT2 plays an important role in regulating metabolic and energetic signals in the diabetic kidney, specifically in KPTCs. Nullification of KPTC-GLUT2 restores, to some extent, the deranged metabolic program of the diabetic kidney and protects it from the development and progression of DKD. However, enhanced mTORC1 activity and subsequently oxidative stress, may alter kidney function in a more chronic phase (Figure 6). Adding novel information regarding the exact mechanisms by which KPTC-GLUT2 functions under diabetic conditions has the potential to be translated into relevant therapeutics for treating DKD in the future.

## 7. Materials and Methods

### 7.1. Animals

The Institutional Animal Care and Use Committee of the Hebrew University (AAALAC accreditation #1285; Ethic approval number MD-19-15784) approved the experimental protocol used. Animal studies are reported, in compliance with the ARRIVE guidelines [122]. To generate diabetic mice lacking GLUT2 in KPTCs, we first crossed mice containing two *loxP* sites flanking the open reading frame of the GLUT2 gene (GLUT2*^fl/fl^*; described in [64]) with the iL1-sglt2-Cre line [123]. Then, KPTC-GLUT2*^fl/fl;sglt2Cre^* (KPTC^GLUT2−/−^) mice were crossed with Akita*^Ins2+/C96Y^* to generate Akita-KPTC^GLUT2−/−^ mice. At three weeks of age, littermates were divided into two groups according to their genotypes: Akita-Cre^−^ and Akita-Cre^+^. Cre^−^ or Cre^+^ refers to the presence or deletion of GLUT2. All animals were homozygous for flox. The mice were monitored weekly for their blood glucose levels and body weight until they were sixteen weeks of age. Then, the mice were euthanized by a cervical dislocation under anesthesia, their kidneys were removed and snap-frozen.

### 7.2. LC-MS Metabolomics Analysis

Frozen kidney samples weighing ~10 mg each were transferred into soft tissue homogenizing tubes containing 1.4 mm ceramic beads (CK14, Bertin corp., Rockville, MD, USA) prefilled with 400 µL of cold (−20 °C) metabolite extraction solvent (methanol:acetonitrile:water, 5:3:2) and kept on ice. Samples were homogenized using Precellys 24 tissue homogenizer cooled to 4 °C (3 × 20 s at 6000 rpm, with a 30 s gap between each of the three cycles, Bertin Technologies, Montigny-le-Bretonneux, France). Homogenized extracts were centrifuged at 18,000× *g* for 15 min at 4 °C, supernatants were collected in microcentrifuge tubes and centrifuged again at 18,000× *g* for 10 min at 4 °C. The supernatants were transferred to glass HPLC vials and kept at 75 °C prior to LC-MS analysis.

LC-MS metabolomic analysis was performed as described previously [124]. Briefly, a Dionex Ultimate 3000 high-performance liquid chromatography (UPLC) system coupled to a Orbitrap Q-Exactive Mass Spectrometer (Thermo Fisher Scientific) was used with a resolution of 35,000 at 200 mass/charge ratio (*m/z*), electrospray ionization and polarity switching mode to enable both positive and negative ions across a mass range of 67 to 1000 *m/z*. The UPLC setup consisted of a ZIC-pHILIC column (SeQuant; 150 mm × 2.1 mm, 5 μm; Merck, Darmstadt, Germany) with a ZIC-pHILIC guard column (SeQuant; 20 mm × 2.1 mm). A total of 5 µL of the kidney extracts were injected and the compounds were separated with a mobile phase gradient of 15 min, starting at 20% aqueous (20 mM ammonium carbonate adjusted to pH 9.2 with 0.1% of 25% ammonium hydroxide) and 80% organic (acetonitrile) and terminated with 20% acetonitrile. The flow rate and column temperature were maintained at 0.2 mL/min and 45 °C, respectively, for a total run time of 27 min. All metabolites were detected using mass accuracy below 5 ppm. Thermo Xcalibur was used for the data acquisition. The analyses were performed with TraceFinder 4.1 (Thermo Fisher Scientific, Waltham, MA, USA) by identified the exact mass of the singly charged ion and by known retention time, using an in-house MS library built by running commercial standards for all detected metabolites. Each identified metabolite intensity was normalized to mg of kidney tissue. Metabolite-Auto Plotter 2 [125] was used for data visualization during data processing.

### 7.3. Real-Time PCR

Total kidney mRNA was extracted using Bio-Tri RNA lysis buffer (Bio-Lab, Israel), followed by DNase I treatment (Thermo Scientific, Rockford, IL, USA), and reverse transcribed using the Iscript cDNA kit (Bio-Rad, Hercules, CA, USA). Real-time PCR was performed using iTaq Universal SYBR Green Supermix (Bio-Rad, Hercules, CA, USA) and the CFX connect ST system (Bio-Rad, Hercules, CA, USA). The primers used to detect mouse genes are listed in Table 2. Mouse genes were normalized to Ubc.

### 7.4. Western Blotting

Kidney homogenates were prepared in a RIPA buffer (25 mM Tris-HCl pH 7.6, 150 mM NaCl, 1% NP-40, 1% sodium deoxycholate, 0.1% SDS). Kidney homogenates were prepared by using the BulletBlender^®^ and zirconium oxide beads (Next Advanced Inc., Raymertown, NY, USA). Protein concentrations were measured with the Pierce™ BCA Protein Assay Kit (Thermo Fisher Scientific, Waltham, MA, USA). Samples were resolved by SDS-PAGE (4–15% acrylamide, 150 V) and transferred to PVDF membranes using the Trans-Blot^®^ Turbo™ Transfer System (Bio-Rad, Hercules, CA, USA). Membranes were then incubated for 1 h in 5% milk (in 1× TBS-T) to block unspecific binding. Membranes were incubated overnight with Rabbit anti-pAMPK (#2535, Cell Signaling, 1:1000), Rabbit anti-CB_1_R (#301214, Immunogen, 1:500), and Rabbit anti-phosphorylated-S6 ribosomal protein (#5364, Cell Signaling, 1:50,000) antibodies at 4 °C. Anti-Rabbit horseradish peroxidase (HRP)-conjugated secondary antibodies (#ab97085, Abcam, 1:2500) were used for 1 h at room temperature, followed by chemiluminescence detection using Clarity™ Western ECL Blotting Substrate (Bio-Rad, Hercules, CA, USA); blot imaging was done using the ChemiDoc™ Touch Imaging System (Bio-Rad, Hercules, CA, USA). Densitometry was quantified using Bio-Rad CFX Manager software. Quantification was normalized to Mouse anti-β actin antibody (#ab49900, Abcam, Cambridge, UK, 1:30,000). Phosphorylated S6 and AMPK were normalized to total Rabbit anti-S6 ribosomal protein (#2217, Cell Signaling, Leiden, The Netherlands, 1:500) and Rabbit anti-total AMPK (#5832, Cell Signaling, Leiden, The Netherlands, 1:500), respectively. The same membranes were washed and re-incubated with antibodies against the normalizing protein and internal controls.

### 7.5. Fluorescence Immunohistochemistry

Kidney sections were deparaffinized and hydrated. Heat-mediated antigen retrieval was performed with 10 mM citrate buffer pH 6.0 (Thermo Scientific, Waltham, MA, USA). Unspecific antigens were blocked by incubating sections for 1 h with 2.5% horse serum (VE-S-2012-50, Vector Laboratories, Newark, CA, USA) and 0.25% Triton X. The sections were stained with a Mouse anti-CPT1A (ab128568, Abcam, Cambridge, UK, 1:500) antibody, followed by incubation with a Goat anti-Mouse-AF488 antibody (ab150117, Abcam, Cambridge, UK, 1:500). Sections were mounted with a mounting medium with DAPI (H-1200, Vector Laboratories, Newark, CA, USA) and photographed using the LSM 700 imaging system (Zeiss, Oberkochen, Germany). The relative fluorescent intensity (RFI) was measured using the ImageJ software (NIH, Bethesda, MD, USA).

### 7.6. Sample Preparation and Endocannabinoid Measurements by LC-MS/MS

Endocannabinoids were extracted, purified, and quantified from kidney lysates. In brief, kidneys were added with ice-cold Tris Buffer, homogenized using the BulletBlender^®^ and zirconium oxide beads (Next Advanced Inc., New York, NY, USA); the protein concentration was determined by the BCA assay. Samples were then supplemented with an ice-cold extraction buffer [1:1 methanol/Tris buffer + an internal standard (IS)] and chloroform/methanol (2:1), vortexed, and centrifuged. The lower organic phase was transferred into borosilicate tubes; this step was repeated three times by adding ice-cold chloroform to the samples and transferring the lower organic phase into the same borosilicate tubes. The samples were dried and kept overnight at −80 °C, then reconstituted with ice-cold chloroform and acetone, kept at −20 °C for 30 min, and then centrifuged to precipitate proteins. Next, the supernatant was dried and reconstituted in an ice-cold LC/MS grade methanol and analyzed on an Sciex (Framingham, MA, USA) QTRAP^®^ 6500^+^ mass spectrometer coupled with a Shimadzu (Kyoto, Japan) UHPLC System. Liquid chromatographic separation was achieved using 5 μL injections of samples onto a Kinetex 2.6 µm C18 (100 × 2.1 mm) column from Phenomenex (Torrance, CA, USA). The autosampler was set at 4 °C and the column was maintained at 40 °C during the entire analysis. The gradient elution mobile phases consisted of 0.1% formic acid in water (phase A) and 0.1% formic acid in acetonitrile (phase B). Endocannabinoids were detected in a positive ion mode using electron spray ionization (ESI) and the multiple reaction monitoring (MRM) mode of acquisition, using d_4_-AEA as IS. The collision energy (CE), declustering potential (DP), and the collision cell exit potential (CXP) for the monitored transitions are presented in Table 3. The levels of AEA and 2-AG in the samples were measured against standard curves, and normalized to the kidney weight.

### 7.7. Fatty Acid Amide Hydrolase Activity Assay

FAAH activity in kidney lysate was determined according to the manufacturer’s instructions (ab252895, Abcam).

### 7.8. Spatial Transcriptomics Data

Spatial transcriptomics of an adult mouse kidney was acquired from the free, open to public 10x Genomics dataset (https://www.10xgenomics.com/resources/datasets/adult-mouse-kidney-ffpe-1-standard-1-3-0 (accessed on 6 December 2022)). The spatial gene expression of GLUT2 (*SLC2A2*), SGLT2 (*SLC5A2*), GLUT1 (*SLC2A1*), and SGT1 (*SLC5A1*) were demonstrated using the 10x Genomics Loupe Browser 5.1.0 (https://support.10xgenomics.com/single-cell-gene-expression/software/downloads/latest#loupe (accessed on 6 December 2022, Pleasanton, CA, USA)).

### 7.9. Statistics

Values are expressed as the mean ± SEM. Unpaired Two-tailed Student’s *t*-test was used to determine the differences between the two groups. Nonparametric Mann–Whitney test was used for unnormal distributed data. GraphPad Prism v6 for Windows (San Diego, CA, USA) was used for statistical analyses. Significance was set at *p* < 0.05.

## Figures and Tables

**Figure 1 cells-12-00094-f001:**
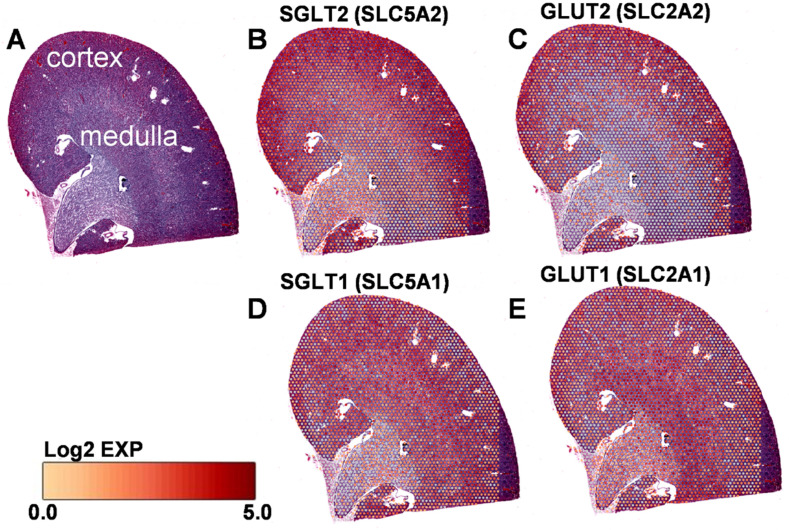
Spatial transcriptomics demonstrates the expression and distribution patterns of the main glucose transporters in the kidney. Regional expression and distribution of different glucose transporters in the mouse kidney correlate with their functionality. Histological snapshot of the kidney (**A**). SGLT2 (*SLC5A2*) (**B**), and GLUT2 (*SLC2A2*) (**C**) are most abundantly expressed in the cortical part of the kidney. SGLT1 (*SLC5A1*) (**D**) and GLUT1 (*SLC2A1*) (**E**) are most abundantly expressed in the corticomedullary/medullar part of the kidney. The scale bar is presented as a log2 scale of each gene accordingly. Adopted from 10× genomics Dataset of mouse kidney. The pictures were generated using Loupe Browser 5.1.0.

**Figure 2 cells-12-00094-f002:**
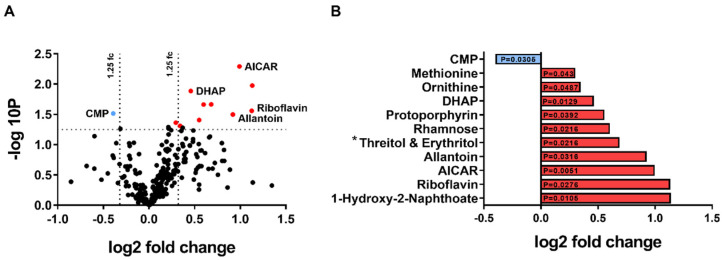
Kidney metabolomic analysis of KPTC-GLUT2 null Akita diabetic mice. We utilized our KPTC-GLUT2 null Akita diabetic mouse model and performed an unbiased metabolomic analysis on their kidney lysate. The volcano plot shows 11 metabolites that were found to be significantly changed in Akita-KPTC^GLUT2−/−^ mice (n = 11), compared with Akita-KPTC^GLUT2+/+^ WT littermates (n = 10) (**A**), the fold change and *p* value for the significantly changed metabolites (**B**). AICAR, 5-Aminoimidazole-4-Carboxamide Ribonucleotide; DHAP, Dihydroxyacetone Phosphate; CMP, Cytidine Monophosphate; * Threitol and Erythritol are isobars, which do not separate chromatographically in our method.

**Figure 3 cells-12-00094-f003:**
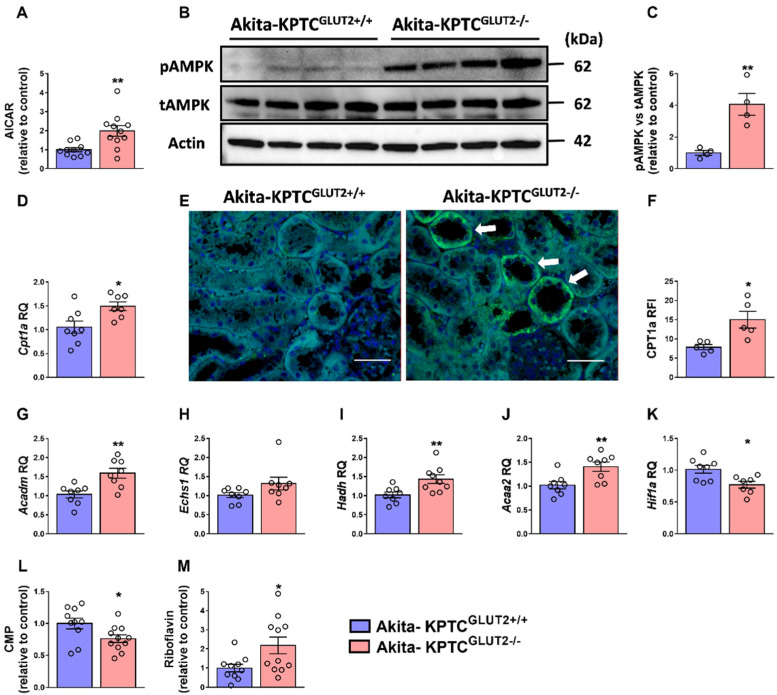
AICAR activates AMPK and restores fat oxidation in KPTC-GLUT2 null Akita diabetic mice. Significant AICAR upregulation in Akita-KPTC^GLUT2−/−^ mice (n = 11), compared with Akita-KPTC^GLUT2+/+^ mice (n = 10) (**A**), was associated with the elevated protein expression levels of pAMPK, shown in a representative immunoblot (**B**) and quantification (n = 4 mice for each group) (**C**). A significant increase in the fatty acid β-oxidation rate-limiting enzyme *Cpt1a* at the mRNA (n = 8 for Akita-KPTC^GLUT2+/+^, n = 7 for Akita-KPTC^GLUT2−/−^) (**D**) and protein levels shown in a representative IF (20× magnification, scale bar: 50 µm) (**E**) and quantification (n = 5 mice for each group) (**F**). The mRNA expression levels of β-oxidation enzymes (n = 8 mice for each group) (**G**–**J**). Significant decreases in the expression of *Hif1a* (**K**), and the levels of CMP (**L**) and increased riboflavin (**M**) metabolite levels (n = 10 for Akita-KPTC^GLUT2+/+^, n = 11 for Akita-KPTC^GLUT2−/−^). The data represent the mean ± SEM and were analyzed by Unpaired Two-tailed Student’s *t*-test; Nonparametric Mann–Whitney test was used for unnormal distributed data. * *p* < 0.05, ** *p* < 0.01. AICAR, 5-Aminoimidazole-4-Carboxamide Ribonucleotide; AMPK oxidative stress, AMP-dependent protein kinase; *Cpt1*, carnitine palmitoyltransferase I; *Acadm*, acyl-CoA dehydrogenase medium chain; *Echs*, enoyl-CoA hydratase; *Hadh*, hydroxyacyl-CoA dehydrogenase; *Acaa2*, acetyl-CoA acyltransferase 2; *Hif1a*, hypoxia-inducible factor-1 alpha; CMP, cytidine monophosphate.

**Figure 4 cells-12-00094-f004:**
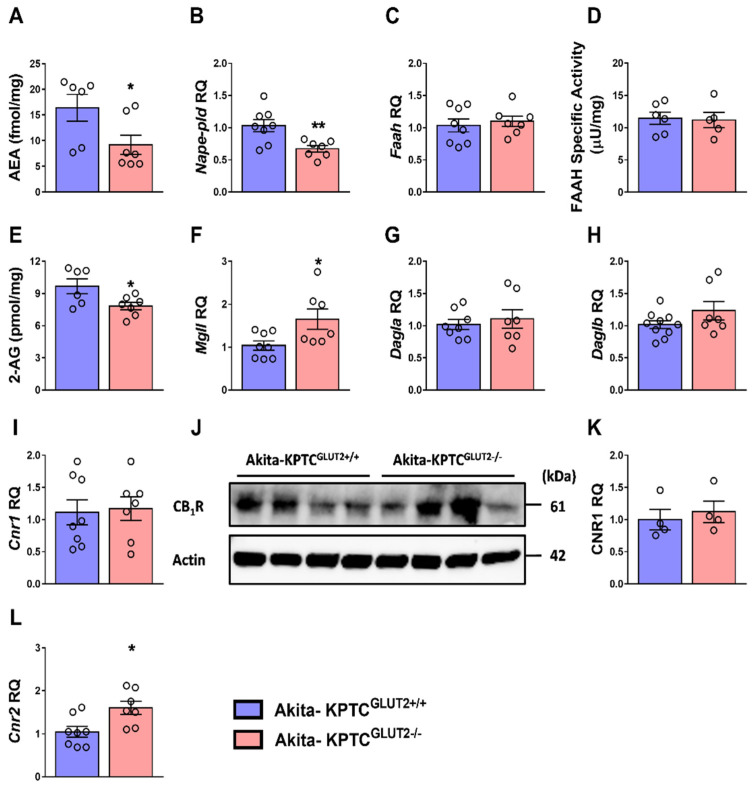
Reduced endocannabinoid ‘tone’ in Akita-KPTC^GLUT2−/−^ mice. Decreased levels of LC-MS/MS measured kidney endocannabinoids, AEA and 2-AG, from Akita-KPTC^GLUT2−/−^ mice, compared to Akita-KPTC^GLUT2+/+^ WT animals (**A**,**E**), corresponding to the decreased kidney *Nape-pld* (**B**) and no change in *Faah* mRNA levels (**C**), and FAAH activity (**D**) as well as increased *Mgll* (**F**) and no changes in *Dagla/b* mRNA levels (**G**,**H**). Kidney CB_1_R (*Cnr1*) mRNA (**I**) and protein levels are shown in a representative immunoblot (**J**) and quantification (**K**). Kidney CB_2_R (*Cnr2*) mRNA expression levels (**L**). For A and E, n = 6 mice for the Akita-KPTC^GLUT2+/+^ group and n = 7 mice for the Akita-KPTC^GLUT2−/−^ group. For B, F–I, and L, n = 8 mice for the Akita-KPTC^GLUT2+/+^ group and n = 7 mice for the Akita-KPTC^GLUT2−/−^ group. For K, n = 4 mice for each group. Data represent the mean ± SEM and were analyzed by Unpaired Two-tailed Student’s *t*-test; Nonparametric Mann–Whitney test was used for unnormal distributed data. * *p* < 0.05, ** *p* < 0.01. AEA, Anandamide; 2-AG, 2-arachidonoylglycerol; *Napepld*, N-acyl phosphatidylethanolamine phospholipase D; *Mgll*, monoacylglycerol lipase; *Cnr*, cannabinoid receptor.

**Figure 5 cells-12-00094-f005:**
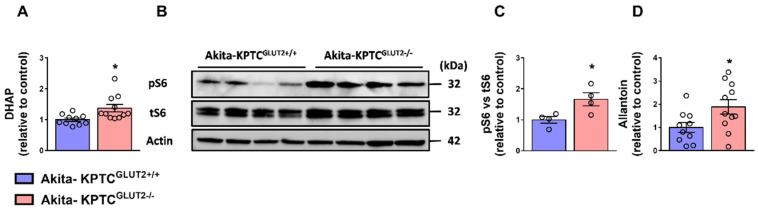
DHAP upregulation and mTORC1 activation in Akita-KPTC^GLUT2−/−^ mice. Significant DHAP upregulation in Akita-KPTC^GLUT2−/−^ mice (n = 11), compared with Akita-KPTC^GLUT2+/+^ mice (n = 10) (**A**). Enhanced mTORC1 activation in Akita-KPTC^GLUT2−/−^ kidneys, manifested by the elevated protein expression levels of pS6, shown in a representative immunoblot (**B**) and quantification (n = 4 mice for each group) (**C**). Significant allantoin upregulation, an oxidative stress biomarker, in Akita-KPTC^GLUT2−/−^ mice (n = 11), compared with Akita-KPTC^GLUT2+/+^ mice (n = 10) (**D**). Data represents the mean ± SEM and were analyzed by Unpaired Two-tailed Student’s *t*-test. * *p* < 0.05. DHAP, Dihydroxyacetone Phosphate.

**Figure 6 cells-12-00094-f006:**
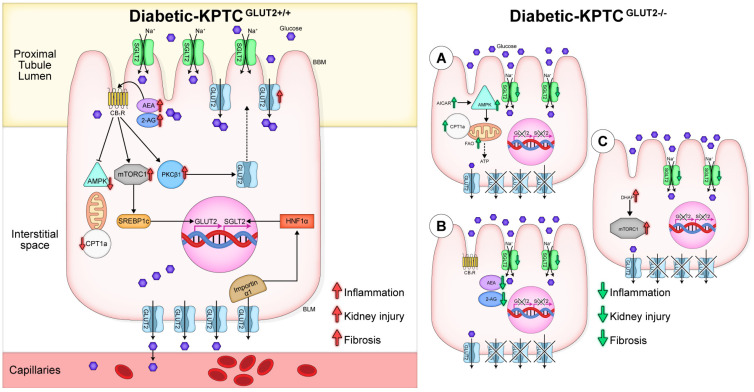
Suggested mechanisms by which GLUT2 nullification in KPTCs affects the development of DKD. Under diabetic conditions, enhanced GLUT2 expression and its apical translocation from the basolateral membrane (BLM) to the apical brush border membrane (BBM) along with elevated SGLT2 expression result in cellular glucotoxicity, increased endocannabinoid ‘tone’, and overactivation of mTORC1. All together, these effects lead to reduced fatty acid oxidation (FAO), and consequently to kidney injury, inflammation, and fibrosis (**left** panel). KPTC-GLUT2 nullification results in reduced SGLT2 expression, decreased glucose absorption, glycosuria, and restored FAO via activation of AMPK/CPT1a signaling (**A**). Moreover, KPTC-GLUT2 nullification reduces cellular endocannabinoid/CB_1_R signaling (**B**); however, it may lead to increased levels of DHAP that activates mTORC1 signaling, which may alter kidney function in a late phase (**C**).

**Table 1 cells-12-00094-t001:** KPTC-GLUT2 expression under hyperglycemia.

Model	Animal	Protein Levels	mRNA Levels	Used Method for Evaluation	Reference
Acute STZ-induced Diabetes	Rat	N/A	Slightly increased	qISH	[42]
Chronic STZ-induced Diabetes	Rat	N/A	5-fold increase	qISH	[42]
2/3/4 weeks uncontrolled STZ-induced Diabetes	Rat (isolated PTs)	increased	increased	NB, WB	[43]
Alloxan-induced Diabetes	Rat	N/A	Two-fold increase	NB	[40]
Alloxan-induced Diabetes (10/20 days)	Rat	increased	increased	NB, WB	[44]
Alloxan-induced Diabetes (with glucosuria)	Rat	~100% increase	~100% increase	NB, WB	[45]
Zucker Diabetic rats	Rat (isolated PTs)	increased	increased	WB, Slot-blot	[46]
STZ-induced DM	Rat	increased	increased	Immuno-blot, NB	[47]
HEPTECs isolated from type-2-Diabetic patients	Human	increased	increased	qPCR, WB	[48]
STZ-induced Diabetes	Rat	increased	N/A	WB, IHC	[49]
STZ-induced Diabetes	Rat	increased	N/A	WB	[50]
STZ-induced Diabetes	Mouse	increased	N/A	IHC	[36]
Diabetic human KPTCs	Human	increased	N/A	IHC	[36]
Akita Diabetic mice	Mouse	increased	increased	IHC, WB, RT-PCR	[36]
Goto-Kakizaki (GK)T2D rats	Rat (purified renal BBM)	increased	N/A	WB	[51]
STZ-induced T1D rats	Rat (purified renal BBM)	increased	N/A	WB	[51]
HFD/JFD-induced insulin resistance	Rat (purified renal BBM)	increased	N/A	WB	[51]
db/db mice (T2D)	Mouse	N/A	increased	RT-PCR	[52]
Renal tissue of T2D patients with preserved kidney function, (unilateral nephrectomy for renal carcinoma)	Human	N/A	Slightly reduced	RT-PCR	[53]
KPTCs treated with HG	Human	increased	N/A	WB, IF	[37]

STZ, streptozotocin; qISH, quantitative in situ hybridization; NB, northern-blot; WB, western-blot; D, diabetes; HEPTECs, human exfoliated proximal tubular epithelial cells from fresh urine; T2D, type 2 diabetes; T1D, type 1 diabetes; HFD, High-fat diet; GK, Goto-Kakizaki (Goto et al., 1976); KPTCs, kidney proximal tubular cells; HG, high glucose (30 mM).

**Table 2 cells-12-00094-t002:** Mouse Primers used for RT-PCR Analysis.

Gene	Forward Primer (5′–3′)	Reverse Primer (5′–3′)
*Acaa2*	CAGAGGTGGAAAGCTGCTAA	GCATGGTCTGTTTGCCTTTC
*Acadm*	CAGCCAATGATGTGTGCTTAC	CATACTCGTCACCCTTCTTCTC
*Cnr1*	AAGTCGATCTTAGACGGCCTT	TCCTAATTTGGATGCCATGTCTC
*Cnr2*	CTGCAGCTCTTGGGACCTAC	TGTCCCAGAAGACTGGGTGT
*Cpt1a*	CCGTGAGGAACTCAAACCTATT	CAGGGATGCGGGAAGTATTG
*Dagla*	GTCCTGCCAGCTATCTTCCTC	CGTGTGGGTTATAGACCAAGC
*Daglb*	AGCGACGACTTGGTGTTCC	GCTGAGCAAGACTCCACCG
*Echs1*	GGACTGTTACTCCAGCAAGTTC	CCCACCAAGAGCATAACCATT
*Faah*	GTATCGCCAGTCCGTCATTG	GCCTATACCCTTTTTCATGCCC
*Hadh*	CCAAGAAGGGAATTGAGGAGAG	ACAAACTCATCTCCAGCCTTAG
*Hif1a*	TTGCTTTGATGTGGATAGCGATA	CATACTTGGAGGGCTTGGAGAAT
*Mgll*	ACCATGCTGTGATGCTCTCTG	CAAACGCCTCGGGGATAACC
*Napepld*	ACGTCCTCCTCTAGTCTGTAATC	AGCGCCAAGCTATCAGTATCC
*Ubc*	GCCCAGTGTTACCACCAAGA	CCCATCACACCCAAGAACA

*Acaa,* acetyl-CoA acyltransferase; *Acadm,* acyl-CoA dehydrogenase medium chain; *Aldo*, aldolase fructose-bisphosphate; *Cnr*, cannabinoid receptor; *Cpt*, diacylglycerol cholinephosphotransferase; *Dagl*, diacylglycerol lipase; *Echs*, enoyl-CoA hydratase; *Faah*, fatty acid amide hydrolase; *Hadh*, hydroxyacyl-CoA dehydrogenase; *Hif*, hypoxia inducible factor; *Mgll*, monoacylglycerol lipase; *Napepld*, N-acyl phosphatidylethanolamine phospholipase D; *Ubc*, ubiquitin C.

**Table 3 cells-12-00094-t003:** MRM transitions for eCB measurements in ESI^+^.

Analyte	Molecular Ion [M + H]^+^ [*m/z*]	Fragment [*m/z*]	DP [Volts]	CE [Volts]	CXP [Volts]
2-AG	379.2	287.1 (quantifier)	70	19	14
		91 (qualifier)	70	67	10
AEA	348.2	287.1 (quantifier)	26	13	16
		62 (qualifier)	26	13	8
d_4_-AEA	352.3	287.1 (quantifier)	66	15	20
		66 (qualifier)	66	21	8

2-arachidonoylglycerol, 2-AG; Anandamide, AEA.

## Data Availability

Not applicable.

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
