# Peer review of "Kidney Proximal Tubule GLUT2—More than Meets the Eye"

_cells, 2022, doi:10.3390/cells12010094_

Round 1

Reviewer 1 Report

The authors in their manuscript (Review + Original Article) entitled, 'Kidney Proximal Tubule GLUT2 - More than Meets the Eye' did a great job in comprehensively presenting and discussing the literature about the role of renal GLUT2 in pathogenesis of metabolic disorders including diabetic kidney disease.

1) The review + new data are timely and relevant to the field of renal GLUT2 and diabetic kidney disease. This is an outstanding manuscript.

2) In 'KPTC-GLUT2: is it targetable?' sub-section, I would like to suggest the authors to include some information about targeting renal GLUT2 in vivo using kidney-specific drug delivery system. There are many recent studies / reviews that present new drug delivery systems specifically targeting the kidneys. Such systems will likely minimize the effects of GLUT2 inhibition on the pancreas, intestine and other tissues. 

3) In reporting of the western blots, I would like to suggest the authors to mention in the methods whether the same membranes / blots were washed and re-probed for different proteins including the internal control. This is important for transparency and reproducibility of the results. Some of the information is already covered in the full images submitted by the authors.

4) In fig. 3E, staining for Cpt1a is unclear. Arrows pointing to the Cpt1a staining will be helpful. Moreover, the nuclear staining is also not clearly visible. Please also mention the length of the scale bar reported in this figure.

Author Response

The authors in their manuscript (Review + Original Article) entitled, 'Kidney Proximal Tubule GLUT2 - More than Meets the Eye' did a great job in comprehensively presenting and discussing the literature about the role of renal GLUT2 in pathogenesis of metabolic disorders including diabetic kidney disease.

1) The review + new data are timely and relevant to the field of renal GLUT2 and diabetic kidney disease. This is an outstanding manuscript.

We would like to deeply thank this reviewer for his/her kind words and for finding our manuscript informative and timely relevant.

2) In 'KPTC-GLUT2: is it targetable?' sub-section, I would like to suggest the authors to include some information about targeting renal GLUT2 in vivo using kidney-specific drug delivery system. There are many recent studies / reviews that present new drug delivery systems specifically targeting the kidneys. Such systems will likely minimize the effects of GLUT2 inhibition on the pancreas, intestine and other tissues.

Thank you for this comment. We included a paragraph describing kidney-specific drug delivery system into this section (please see Page 18, lines 652-660).

3) In reporting of the western blots, I would like to suggest the authors to mention in the methods whether the same membranes / blots were washed and re-probed for different proteins including the internal control. This is important for transparency and reproducibility of the results. Some of the information is already covered in the full images submitted by the authors.

Amended in the methods (please see Page 21, lines 780-782).

4) In fig. 3E, staining for Cpt1a is unclear. Arrows pointing to the Cpt1a staining will be helpful. Moreover, the nuclear staining is also not clearly visible. Please also mention the length of the scale bar reported in this figure.

The representative images were amended and clearly point to the CPT1a expression (please see Page 13, Fig. 3E and line 482 in the legend).  

Reviewer 2 Report

The review deals with a subject of interest such as the functional role of the GLUT2 transporters of the proximal tubule cells and explores the molecular mechanisms involved in their regulation based on original results and also a correct background review of the authors and extensive to other colleagues who work on the subject.

The manuscript is correctly written and a series of comments have been made in the attached pdf file in order to achieve the best version of the paper for publication.

The general recommendation would perhaps be to put the participation of SGL1 and GLUT1 in the transport of glucose in the proximal tubule in context at the beginning of the text, and then focus on the focus of the work, SGLT2 and GLUT2.

Finally, in the specific recommendations made in the highlighted pdf archive, one can perhaps observe the need to put in a separate section all the theoretical gaps that are mentioned throughout the text and that perhaps are lost in the large number of mechanisms and models that are described correctly throughout the text.

I hope that the comments are well received by the authors as they seek to make the review of interest to the general scientific community.

Author Response

The review deals with a subject of interest such as the functional role of the GLUT2 transporters of the proximal tubule cells and explores the molecular mechanisms involved in their regulation based on original results and also a correct background review of the authors and extensive to other colleagues who work on the subject.

The manuscript is correctly written and a series of comments have been made in the attached pdf file in order to achieve the best version of the paper for publication.

The general recommendation would perhaps be to put the participation of SGL1 and GLUT1 in the transport of glucose in the proximal tubule in context at the beginning of the text, and then focus on the focus of the work, SGLT2 and GLUT2.

Finally, in the specific recommendations made in the highlighted pdf archive, one can perhaps observe the need to put in a separate section all the theoretical gaps that are mentioned throughout the text and that perhaps are lost in the large number of mechanisms and models that are described correctly throughout the text.

I hope that the comments are well received by the authors as they seek to make the review of interest to the general scientific community.

We would like to greatly thank this reviewer for his valuable comments and recommendations. The entire manuscript and figures were amended accordingly. We also added the requested section related to the theoretical gaps. Please find our point-by-point response to all of the comments in the attached PDF file as well as in the “MARKED” manuscript.
